# OpenReview forum: "CamoDocs: Poisoning Attack against Retrieval-Augmented Language Models"
_ICLR.cc/2026/Conference — Submitted to ICLR 2026_

### Official Review · Reviewer_e2hp · 2025-10-27

**Soundness:** 2
**Presentation:** 3
**Contribution:** 2
**Rating:** 4
**Confidence:** 4

**Summary:**

This paper propose CamoDocs for poisoning the RAG system. The method works in two stages. First, it crafts sub-documents by generating adversarial content and retrieving relevant benign content. Second, it crafts the final adversarial documents by performing token-level manipulation on the benign sub-documents to disperse their embeddings and merging these camouflaged chunks with the adversarial chunks. Authors evaluate this attack against several LLMs and datasets, claiming a high ASR against heuristic defenses and a "non-trivial" ASR of ~27% against the TrustRAG defense.

**Strengths:**

- The paper addresses the security of RAG systems, which is important.

- The authors correctly identify a clear and trivial-to-exploit flaw in the baseline PoisonedRAG attack (its reliance on query prepending).

- The two-stage procedure is described with sufficient clarity to be understood.

**Weaknesses:**

- RQ2 and Table 2 prove that this paper's core methodology (using retrieved benign documents) is inferior to a simpler variant (using synthesized benign documents). This invalidates the paper's central claims.

- The paper describes a 27.78% ASR as "intolerable." This is a overstatement.

- The victim retriever is outdated. It is unclear if these vulnerabilities exist in SOTA retrievers having more robust embedding spaces.

- The loss function merely disperses embeddings from their centroid. It might fool k-means, but it is unlikely to fool more robust density-based clustering (DBSCAN) or outlier detection algorithms. This was not tested.

- No adaptive defense is considered.

- missing references

[1]Machine Against the RAG: Jamming Retrieval-Augmented Generation with Blocker Documents.

[2]Enhancing Noise Robustness of Retrieval-Augmented Language Models with Adaptive Adversarial Training

[3]Understanding Data Poisoning Attacks for RAG: Insights and Algorithms

[4]Trojanrag: Retrieval-augmented generation can be backdoor driver in large language models

[5]Towards More Robust Retrieval-Augmented Generation: Evaluating RAG Under Adversarial Poisoning Attacks

**Questions:**

1. See weakness
2. Can you justify the claim that a 27.78% ASR is "intolerable"? This implies the TrustRAG defense is ~72% effective at stopping your attack. Why do you consider this a "failure" for the defense rather than a "failure" for the attack?

---

> ### Author Response · Authors · 2025-12-03
>
> ### **Response regarding the Synthesized vs. Retrieved Benign Documents**
>
> Thank you for the insightful comment. We respectfully argue that using synthesized benign documents (CamoDocs + GPT-4) in Table 2 is also an instance of our methodology, and this result does not invalidate our core claims. Our paper’s central claim is that the CamoDocs pipeline, which combines sub-document merging and token manipulation, can bypass existing defense mechanisms, thereby demonstrating the vulnerability of current RAG systems. The CamoDocs + GPT-4 variant utilizes this exact pipeline. Far from invalidating our central claim, this result demonstrates the robustness of CamoDocs, as it achieves high Attack Success Rates (ASR) in both settings: using synthesized benign documents and using documents retrieved by a surrogate retriever. Ultimately, the source of the benign documents used for token manipulation depends on the attacker’s capability.
>
>
> ### **Justification for the "Intolerable" ASR Claim**
>
> Thank you for the insightful comment. In the AI security field, especially regarding adversarial attacks, the main concern for research is vulnerability in critical domains such as healthcare and finance. In these critical domains, high reliability is required. For example, there is a real-world example in which a retrieval corruption attack incurred a $2.5K loss when ChatGPT generated code that contained a malicious code snippet. In addition, we can imagine a situation where an LLM is adopted in the healthcare field; if the LLM gives wrong information to doctors and patients, the consequence is intolerable. Given these disastrous consequences and the expected wide adoption of LLMs in many applications, we believe a 27.78% ASR is still intolerable, especially in fields where high reliability is required. In addition, we can achieve a higher ASR with some variants. We additionally tested increasing the poisoning ratio up to 0.029% (which is still very small compared to 0.009%), and it achieved 31.20% with Llama-3.1-8B on HotpotQA.
>
> ### **Evaluation on State-of-the-Art (SOTA) Retrievers**
>
> We appreciate the reviewer's suggestion to verify our method against more recent, robust retrievers. To address this, we conducted additional experiments using the **Qwen3-Embedding-0.6B** retriever, which currently holds the top rank on the MTEB leaderboard among models of similar size.
>
> As shown in the table below, CamoDocs maintains a significantly higher Attack Success Rate (ASR) compared to baselines, even though the SOTA retriever exhibits higher clean accuracy due to its superior capability (44.40% for Qwen3 vs. 40.50% for Contriever used in the main paper). This result confirms that transferability among retriever models exists, and CamoDocs effectively leverages this property to remain effective against SOTA systems. The results below were obtained using Llama-3.1-8B on HotpotQA.
>
> | Defense | Attack | ASR | ACC |
> | :--- | :--- | :--- | :--- |
> | **No attack** | - | - | 44.40 |
> | **Query Detection** | PoisonedRAG | 7.20 | 27.60 |
> | | PIA | 3.50 | 44.30 |
> | | CamoDocs | 53.20 | 24.70 |
> | **TrustRAG** | PoisonedRAG | 6.40 | 32.40 |
> | | PIA | 7.50 | 40.00 |
> | | CamoDocs | 20.20 | 37.50 |

---

> ### Author Response · Authors · 2025-12-03
>
> ### **Testing against Density-Based Clustering Defense (DBSCAN)**
>
> Thank you for your insightful comment regarding the potential robustness of density-based clustering against our method. To address this, we evaluated CamoDocs against a defense mechanism utilizing DBSCAN and found that our attack remains highly effective, achieving a significantly higher Attack Success Rate (ASR) compared to the K-means defense.
>
> Unlike K-means, which forces data into a pre-specified number of clusters (e.g., $k=2$, allowing the defense to attempt to separate benign from adversarial), DBSCAN relies on a density radius ($\epsilon$) and does not guarantee a fixed number of clusters. This structural difference actually renders DBSCAN more vulnerable to CamoDocs, as our method disperses adversarial embeddings to resemble background noise or merge with benign distributions.
>
> The effectiveness of DBSCAN is highly sensitive to the $\epsilon$ parameter, leading to two failure modes against CamoDocs:
>
> 1.  **Small $\epsilon$ (Over-segmentation):** If $\epsilon$ is too small, the algorithm fails to find sufficient neighbors for any point. Consequently, almost all embeddings are classified as "noise" (outliers) rather than forming a cohesive cluster. Since the defense relies on identifying a "tight adversarial cluster" to filter, it fails to act.
> 2.  **Large $\epsilon$ (Under-segmentation):** If $\epsilon$ is too large, the algorithm groups benign and adversarial documents into a single, massive cluster. Due to the dispersion of our adversarial embeddings, this merged cluster exhibits low average cosine similarity, preventing the defense from flagging it as a suspicious, concentrated attack cluster.
>
> We demonstrate this behavior in the table below using Llama-3.1-8B on HotpotQA. We report the **Noisy Label Proportion** (the fraction of retrieved documents classified as noise/outliers) and **Average Cosine Similarity** (the average intra-cluster similarity).
>
> | Epsilon | Noisy Label Proportion | Average Cosine Similarity | Attack Success Rate | Clean Accuracy |
> | :--- | :--- | :--- | :--- | :--- |
> | 0.50 | 1.000 | - | 45.50 | 31.40 |
> | 0.60 | 1.000 | - | 45.20 | 31.60 |
> | 0.70 | 0.999 | 0.325 | 45.50 | 31.80 |
> | 0.80 | 0.995 | 0.264 | 46.50 | 31.20 |
> | 0.90 | 0.977 | 0.161 | 44.00 | 31.50 |
> | 1.00 | 0.920 | 0.068 | 45.50 | 31.40 |
>
> As shown in the table:
> * At **$\epsilon \le 0.60$**, the noisy label proportion is 100%, meaning DBSCAN failed to form any clusters.
> * At **$\epsilon \ge 0.70$**, clusters begin to form, but the average cosine similarity drops precipitously (from 0.325 to 0.068) as $\epsilon$ increases. This indicates that the formed clusters are sparse and diverse, not the "tight" clusters that defenses target.
>
> Because the clusters formed are either non-existent or too sparse to trigger a threshold-based detection, no documents are filtered. Consequently, CamoDocs achieves an ASR between **44.0% and 46.5%** against DBSCAN. This is notably higher than the ASR against K-means (29.40%), confirming that dispersion makes density-based detection harder, not easier, for the defender.

---

> ### Author Response · Authors · 2025-12-03
>
> ### **Comparison with Recent Attack and Defense Baselines**
>
> Thank you for the insightful comment. We have expanded our evaluation to include the **CorruptRAG** [6] attack baseline and the **Divide-and-Vote** [7] defense baseline.
>
> **CorruptRAG** operates by injecting an adversarial document which claims that the corpus containing the correct answer is outdated, asserting the attacker’s target answer is the correct one.
> **Divide-and-Vote** [7] is a defense strategy that generates a prediction for each retrieved document individually and aggregates the results via majority voting.
>
> As shown in the table below, our experiments yielded the following insights:
>
> 1.  **Effectiveness against Query Detection:** CorruptRAG is ineffective against the Query Detection defense (ASR: 4.90%). This is because, similar to PoisonedRAG, CorruptRAG prepends the target query to the adversarial document to ensure retrieval. This specific characteristic makes it easily detectable by the query-detection mechanism.
>
> 2.  **Effectiveness against Divide-and-Vote:** The attack is also ineffective against Divide-and-Vote (ASR: 8.50%). This defense isolates the poisoned document from the benign documents by feeding them to the LLM separately. Since the LLM generates an output for each document in the top-k results, the single poisoned document affects only one prediction. The remaining benign documents produce correct answers, which dominate the aggregation result, leading to a correct final answer.
>
> 3.  **Effectiveness against TrustRAG:** CorruptRAG achieves a comparatively higher ASR against TrustRAG (34.80%). Because CorruptRAG injects only a single poisoned document, it does not form the dense clusters that TrustRAG's clustering defense relies on for detection. Furthermore, unlike PIA, it does not contain explicit malicious instructions, allowing it to bypass TrustRAG's malicious instruction filtering process.
>
> **Conclusion on Robustness:**
> While CorruptRAG achieves a higher ASR against TrustRAG specifically, it is rendered ineffective (<9% ASR) by the other two defenses. In contrast, CamoDocs demonstrates consistent effectiveness, maintaining a significant threat level (ASR > 29%) across *all* evaluated defenses. This confirms that CamoDocs is a more versatile attack strategy in diverse defense environments.
>
> | Defense | Attack | Attack Success Rate | Clean Accuracy |
> | :--- | :--- | :--- | :--- |
> | **Query Detection** | PoisonedRAG | 7.20 | 26.60 |
> | | PIA | 4.90 | 39.90 |
> | | CorruptRAG | 4.90 | 39.90 |
> | | CamoDocs | 68.90 | 15.40 |
> | **Divide-and-Vote** | PoisonedRAG | 53.50 | 12.00 |
> | | PIA | 8.00 | 27.50 |
> | | CorruptRAG | 8.50 | 28.00 |
> | | CamoDocs | 65.00 | 13.50 |
> | **TrustRAG** | PoisonedRAG | 6.50 | 33.30 |
> | | PIA | 7.60 | 39.20 |
> | | CorruptRAG | 34.80 | 37.90 |
> | | CamoDocs | 29.40 | 32.70 |
>
> [6] Zhang, B., Chen, Y., Fang, M., Liu, Z., Nie, L., Li, T., & Liu, Z. (2025). Practical poisoning attacks against retrieval-augmented generation. *arXiv preprint arXiv:2504.03957*.
>
> [7] Pan, Y., Pan, L., Chen, W., Nakov, P., Kan, M. Y., & Wang, W. (2023). On the Risk of Misinformation Pollution with Large Language Models. *Findings of the Association for Computational Linguistics: EMNLP 2023*.

---

> ### Author Response · Authors · 2025-12-03
>
> ### **Missing References**
>
> Thank you for bringing these relevant references to our attention. We have included all of them in the revised paper.

---

### Official Review · Reviewer_vpWp · 2025-10-31

**Soundness:** 3
**Presentation:** 2
**Contribution:** 2
**Rating:** 2
**Confidence:** 5

**Summary:**

In this paper, the authors introduce CamoDocs, a method that generates adversarial documents by dividing texts into chunks and merging optimized benign sub-documents with adversarial components.

**Strengths:**

1. The paper presents a new attack method designed to compromise RAG systems.

2. Experimental results are provided to demonstrate the effectiveness of the proposed approach.

**Weaknesses:**

1. The runtime cost of the proposed attack and all baseline methods is not evaluated.

2. The paper does not clearly specify how many poisoned documents are injected per query.

3. Several recent attack and defense methods are not included in the comparison.

4. The number of queries used for each dataset is too small to ensure reliable evaluation.

**Questions:**

1. The approach depends heavily on surrogate embedding models (such as ANCE) and surrogate retrievers (like BM25) to approximate the target dense retriever (Contriever). However, the paper does not investigate how sensitive the attack performance is to discrepancies between the surrogate and target retrievers.

2. The attack requires iterative token manipulation and optimization across multiple sub-documents, which is likely to incur high computational cost. Yet, the paper does not include any analysis of runtime, optimization efficiency, or scalability to larger datasets or more queries.

3. The paper does not clearly specify how many poisoned documents are injected per query, leaving the exact attack scale ambiguous.

4. The study omits comparisons with several recent and more sophisticated poisoning attacks on RAG systems, such as [a][b].

5. The evaluated defenses are limited and overly simplistic. More advanced and robust defenses, such as [c][d][e], should be considered to provide a more comprehensive evaluation.

6. Only 50 queries are sampled for each dataset, which is too small to draw statistically reliable or generalizable conclusions.



[a] Practical Poisoning Attacks against Retrieval-Augmented Generation.

[b] FlippedRAG Black-Box Opinion Manipulation Adversarial Attacks to Retrieval-Augmented Generation Models.

[c] Certifiably Robust RAG against Retrieval Corruption.

[d] Traceback of Poisoning Attacks to Retrieval-Augmented Generation.

[e] TrustRAG: Enhancing Robustness and Trustworthiness in RAG.

---

> ### Author Response · Authors · 2025-12-03
>
> ### **Sensitivity to Surrogate-Victim Discrepancies**
>
> Thank you for the insightful comment. To address the concern regarding how discrepancies between the surrogate and victim retrievers affect attack performance, we conducted additional experiments using different surrogate models.
>
> First, it is important to note that our original experimental setup (presented in the main paper) already represents a scenario with **significant discrepancy**: we used BM25 (a sparse retriever) as the surrogate to attack Contriever (a dense retriever). The fact that CamoDocs achieves a substantial Attack Success Rate (ASR) of 29.40% under this setting demonstrates its robustness even when the attacker lacks knowledge of the victim's retrieval architecture.
>
> To explicitly quantify the sensitivity, we evaluated two additional scenarios: (1) using a different dense retriever (Qwen3-Embedding-0.6B) and (2) using the exact same retriever (Contriever) as the surrogate. The results are summarized in the table below.
>
> | Surrogate Retriever | Victim Retriever | Attack Success Rate | Clean Accuracy |
> | :--- | :--- | :--- | :--- |
> | **BM25** (Sparse) | Contriever | 29.40 | 32.70 |
> | **Contriever** (Dense) | Contriever | 35.90 | 32.80 |
> | **Qwen3-Emb-0.6B** (Dense) | Contriever | 32.10 | 33.00 |
>
> **Observations:**
> 1.  **Dense-to-Dense Transferability:** When the surrogate is switched from a sparse model (BM25) to another dense model (Qwen3-Embedding-0.6B), the ASR improves from 29.40% to 32.10%. This suggests that transferability is higher when the surrogate and victim share the same retrieval mechanism (dense retrieval).
> 2.  **Ideal Conditions:** As expected, when the surrogate retriever perfectly matches the victim retriever (Contriever), the ASR reaches its peak at 35.90%, an increase of 6.5 percentage points over the BM25 baseline.
>
> **Conclusion:**
> These results confirm that while minimizing the discrepancy between the surrogate and victim retrievers does enhance the attack success rate, CamoDocs remains highly effective even under significant architectural mismatches (e.g., Sparse vs. Dense). This highlights the practical threat of the attack in black-box scenarios.

---

> ### Author Response · Authors · 2025-12-03
>
> ### **Clarification on Poisoning Ratio and Its Impact on ASR**
>
> Thank you for the insightful and helpful comment.
>
> We used **10 poisoned documents per query** ($\beta = 10$) across all datasets. This is derived from setting **$k = 5$** for the top-k selection of benign documents and a chunk count $\gamma = 2$ ($5 \times 2 = 10$).
>
> As described in **Section 4.1**, we tested 50 queries per repetition. Consequently, the corresponding poisoning ratios are **0.01%**, **0.02%**, and **0.01%** for **HotpotQA**, **NQ**, and **MS-MARCO**, respectively. This poisoning ratio is much less than 1%, ensuring the true attack scale remains very small.
>
> The exact statistics are summarized in the table below. Note that the tested queries do not overlap between different repetitions. This means that testing 50 queries per repetition over 10 repetitions allows us to evaluate **500 queries total**, but with a reduced poisoning ratio. This reduced poisoning ratio occurs because we only inject adversarial documents for the target queries tested in the *current* repetition. This is distinct from a scenario where one tests 500 queries at once with all corresponding poisoned documents injected simultaneously. By injecting adversarial documents only for the 50 queries currently being tested, we maintain a realistic and low poisoning budget per experiment.
>
> | Dataset | Poisoned Docs per Query ($\beta$) | Tested Queries per Repetition | Total Poisoned Docs per Repetition | Corpus Size | Poisoning Ratio |
> | :--- | :--- | :--- | :--- | :--- | :--- |
> | HotpotQA | 10 | 50 | 500 | 5,233,329 | 0.01% |
> | NQ | 10 | 50 | 500 | 2,681,468 | 0.02% |
> | MS-MARCO | 10 | 50 | 500 | 8,841,823 | 0.01% |
>
> In the revised paper, we increased the number of tested queries per repetition from 50 to 100. Consequently, the total poisoned documents per repetition and the poisoning ratio doubled from 0.009%, 0.019%, 0.006% to 0.019%, 0.037%, and 0.011%, respectively.
>
> Regarding the parameter $\beta$, we conducted an additional experiment to observe how it affects the attack success rate. We increased $\beta$ from 10 to 15, 20, and 25 by selecting the Top-K ($K=5$) retrieved documents using the surrogate retriever and setting the chunk count ($\gamma$) to 3, 4, and 5. In other words, we increased $\beta$ by splitting the benign and adversarial sub-documents in a more fine-grained manner. The results below were obtained with Llama-3.1-8B on HotpotQA under the TrustRAG defense.
>
> | $\beta$ | Attack Success Rate | Clean Accuracy | Poisoning Ratio | Proportion of Retrieved Adv. Docs (Before Defense) | Proportion of Retrieved Adv. Docs (After Defense) |
> | :--- | :--- | :--- | :--- | :--- | :--- |
> | 10 | 28.10 | 31.50 | 0.019% | 93.62% | 54.06% |
> | 15 | 31.20 | 32.10 | 0.029% | 93.40% | 65.42% |
> | 20 | 28.80 | 33.50 | 0.038% | 92.90% | 62.00% |
> | 25 | 20.20 | 33.00 | 0.048% | 95.38% | 46.66% |
>
> As seen in the table above, increasing $\beta$ and the poisoning ratio helps increase the attack success rate to some extent; for instance, the attack success rate increases by 3.1 percentage points when $\beta$ increases from 10 to 15. However, the attack success rate does not increase monotonically with $\beta$, as it decreases again starting from $\beta=20$.
>
> This is because, although the increased poisoning ratio helps increase the proportion of retrieved adversarial documents among the top-k retrieved documents, it also aids the K-means clustering defense in detecting them. A large number of retrieved adversarial documents tends to form a tight cluster that is easily detectable by the K-means algorithm. Conversely, a small number of retrieved adversarial documents is less detectable because there are fewer points to form a cluster. This analysis is confirmed by the steep decline in the proportion of retrieved adversarial documents **after** applying the K-means clustering defense as $\beta$ increases to 25 (dropping to 46.66%), even though the proportion **before** defense increases (95.38%).

---

> ### Author Response · Authors · 2025-12-03
>
> ## **Computational cost analysis**
>
> Thank you for the insightful comment. We conducted a computational cost analysis for each stage of our algorithm. We synthesize adversarial documents with the synthesizer LLM (GPT-4o-mini), which is also used in PoisonedRAG, and we use a lightweight word-based sparse retriever (BM25) as a surrogate retriever.
>
> ### **Computational cost for document chunking**
>
> Chunking a document requires a single pass over the document. Let the average document length for benign documents be $L_{bn}$, the average document length for adversarial documents be $L_{adv}$, and let $k$ be the number of retrieved benign documents and synthesized adversarial documents per query.
>
> Then, the computational cost for document chunking is $O(kL_{bn} + kL_{adv})$, since we need one pass over each text for chunking.
>
> ### **Computational cost for token manipulation**
>
> Token manipulation requires one batched forward pass with batch size equal to the final target number of adversarial documents $\beta$, one batched backward pass with the loss $L$ to obtain the gradient $\nabla_{e_t} L$ used for the first-order approximation, and $m$ additional forward passes, where $m$ is the number of candidate tokens that show high approximate loss. After chunking, each document is split into $\gamma$ sub-documents, so the average sub-document length becomes $L_{\text{sub, bn}} = L_{bn} / \gamma$ and $L_{\text{sub, adv}} = L_{adv} / \gamma$ for benign and adversarial sub-documents, respectively.
>
> Let $T_{\text{fwd}}(L_{\text{sub}})$ and $T_{\text{bwd}}(L_{\text{sub}})$ denote the time for a single forward pass and a single backward pass, respectively, of the surrogate encoder $E_{\text{surr}}$ on a sub-document of length $L_{\text{sub}}$. Under the standard assumption that encoder cost scales linearly with batch size, one batched forward pass over the $\beta$ current benign sub-documents costs $\beta T_{\text{fwd}}(L_{\text{sub,bn}})$ and the corresponding batched backward pass costs $\beta T_{\text{bwd}}(L_{\text{sub,bn}})$.
>
> In addition, given the gradient with respect to the chosen token’s embedding, we score at most $m$ candidate token embeddings using inner products $\nabla_{e_t} L \cdot e_t$, which costs $O(md)$ per step, where $d$ is the embedding dimension.
>
> The $m$ candidate evaluations require $m T_{\text{fwd}}(L_{\text{sub,bn}})$.
>
> Therefore, each token-manipulation step has complexity
> $$
> O\big(\beta T_{\text{fwd}}(L_{\text{sub,bn}}) + \beta T_{\text{bwd}}(L_{\text{sub,bn}}) + m T_{\text{fwd}}(L_{\text{sub,bn}}) + md\big),
> $$
> and optimizing for $\alpha$ steps yields a total token-manipulation cost per query of
> $$
> O\big(\alpha [\beta T_{\text{fwd}}(L_{\text{sub,bn}}) + \beta T_{\text{bwd}}(L_{\text{sub,bn}}) + m T_{\text{fwd}}(L_{\text{sub,bn}}) + md]\big).
> $$
>
> Therefore, the overall cost is dominated by the encoder forward and backward passes.
>
> ### **Computational cost for sub-document merging**
>
> After token manipulation, we merge the optimized benign sub-documents with the adversarial sub-documents to form the final adversarial documents.
>
> To form one final adversarial document, we read one optimized benign sub-document of length $L_{bn}/\gamma$
> and one adversarial sub-document of length $L_{adv}/\gamma$, and we write their concatenation, which has length $L_{bn}/\gamma + L_{adv}/\gamma$.
>
> Thus, the total number of token operations (reads and writes) per adversarial document is $2[L_{bn}/\gamma + L_{adv}/\gamma]$.
> Repeating this process for the $\beta$ final adversarial documents per query yields a total sub-document merging cost of $O\big(\beta \cdot [2(L_{bn}/\gamma + L_{adv}/\gamma)]\big)$.
>
> ### **Overall per-query complexity**
>
> Putting these stages together, the computational cost of CamoDocs for a single target query consists of:
>
> - document chunking: $O(kL_{bn} + kL_{adv})$,
> - token manipulation: $O\big(\alpha [\beta T_{\text{fwd}}(L_{\text{sub,bn}}) + \beta T_{\text{bwd}}(L_{\text{sub,bn}}) + m T_{\text{fwd}}(L_{\text{sub,bn}}) + md]\big)$,
> - sub-document merging: $O\big(\beta \cdot [2(L_{bn}/\gamma + L_{adv}/\gamma)]\big)$.
>
> All of these costs are incurred **offline** when constructing the poisoned documents. At inference time, the RAG system with poisoned documents uses the same retriever and LLM as in the clean setting and PoisonedRAG, so the online latency is essentially unchanged.
>
> To generate one adversarial document for one target query, CamoDocs takes about **32 seconds** on a single A6000 GPU with PyTorch 2.7.1 and CUDA 12.6. Token manipulation dominates the runtime because it requires multiple forward passes and one batched backward pass of the embedding model. We randomly selected 100 HotpotQA test queries, measured the time for each procedure, and report the averages.
>
> **Breakdown of CamoDocs runtime**
>
> |Procedure|Average Runtime (seconds)|
> |-|-|
> |Document chunking|0.000|
> |Token manipulation|31.754|
> |Sub-document merging|0.000|

---

> ### Author Response · Authors · 2025-12-03
>
> ### **Comparison with Recent Attack and Defense Baselines**
>
> Thank you for the insightful comment. We have expanded our evaluation to include the **CorruptRAG** [a] attack baseline and the **Divide-and-Vote** [f] defense baseline.
>
> **CorruptRAG** operates by injecting an adversarial document which claims that the corpus containing the correct answer is outdated, asserting the attacker’s target answer is the correct one.
> **Divide-and-Vote** [f] is a defense strategy that generates a prediction for each retrieved document individually and aggregates the results via majority voting.
>
> As shown in the table below, our experiments yielded the following insights:
>
> 1.  **Effectiveness against Query Detection:** CorruptRAG is ineffective against the Query Detection defense (ASR: 4.90%). This is because, similar to PoisonedRAG, CorruptRAG prepends the target query to the adversarial document to ensure retrieval. This specific characteristic makes it easily detectable by the query-detection mechanism.
>
> 2.  **Effectiveness against Divide-and-Vote:** The attack is also ineffective against Divide-and-Vote (ASR: 8.50%). This defense isolates the poisoned document from the benign documents by feeding them to the LLM separately. Since the LLM generates an output for each document in the top-k results, the single poisoned document affects only one prediction. The remaining benign documents produce correct answers, which dominate the aggregation result, leading to a correct final answer.
>
> 3.  **Effectiveness against TrustRAG:** CorruptRAG achieves a comparatively higher ASR against TrustRAG (34.80%). Because CorruptRAG injects only a single poisoned document, it does not form the dense clusters that TrustRAG's clustering defense relies on for detection. Furthermore, unlike PIA, it does not contain explicit malicious instructions, allowing it to bypass TrustRAG's malicious instruction filtering process.
>
> **Conclusion on Robustness:**
> While CorruptRAG achieves a higher ASR against TrustRAG specifically, it is rendered ineffective (<9% ASR) by the other two defenses. In contrast, CamoDocs demonstrates consistent effectiveness, maintaining a significant threat level (ASR > 29%) across *all* evaluated defenses. This confirms that CamoDocs is a more versatile attack strategy in diverse defense environments.
>
> | Defense | Attack | Attack Success Rate | Clean Accuracy |
> | :--- | :--- | :--- | :--- |
> | **Query Detection** | PoisonedRAG | 7.20 | 26.60 |
> | | PIA | 4.90 | 39.90 |
> | | CorruptRAG | 4.90 | 39.90 |
> | | CamoDocs | 68.90 | 15.40 |
> | **Divide-and-Vote** | PoisonedRAG | 53.50 | 12.00 |
> | | PIA | 8.00 | 27.50 |
> | | CorruptRAG | 8.50 | 28.00 |
> | | CamoDocs | 65.00 | 13.50 |
> | **TrustRAG** | PoisonedRAG | 6.50 | 33.30 |
> | | PIA | 7.60 | 39.20 |
> | | CorruptRAG | 34.80 | 37.90 |
> | | CamoDocs | 29.40 | 32.70 |
>
> *[f] Pan, Y., Pan, L., Chen, W., Nakov, P., Kan, M. Y., & Wang, W. (2023). On the Risk of Misinformation Pollution with Large Language Models. Findings of the Association for Computational Linguistics: EMNLP 2023.*

---

> ### Author Response · Authors · 2025-12-03
>
> ### **Clarification on Tested Number of Queries**
>
> Thank you for the comment. We would like to clarify that our original evaluation actually involved testing **500 unique queries**, not just 50. The experiment was designed to test 50 distinct queries per repetition over 10 independent repetitions.
>
> This strategy allows us to evaluate 500 queries total but with a significantly **reduced poisoning ratio**. This reduction occurs because we only inject adversarial documents for the target queries tested in the *current* repetition. This is distinct from a scenario where one tests 500 queries at once with all corresponding poisoned documents injected simultaneously. By injecting documents only for the 50 queries currently being tested, we maintain a realistic and low poisoning budget per experiment. We apologize if this aspect of the experimental design was unclear.
>
> In fact, the number of queries we tested is **five times larger** than that of previous work like PoisonedRAG, which evaluated only 100 unique queries (10 queries per repetition over 10 repetitions). Furthermore, to fully address the concern regarding statistical reliability, we have increased the sample size in the revised paper to **1,000 queries** (100 non-overlapping queries across 10 repetitions) for each dataset. Our results confirm that the conclusion remains unchanged with this larger sample size: CamoDocs consistently achieves a high Attack Success Rate (ASR) across all tested defenses, demonstrating its generalized effectiveness.

---

### Official Review · Reviewer_uH6r · 2025-10-31

**Soundness:** 3
**Presentation:** 3
**Contribution:** 2
**Rating:** 2
**Confidence:** 3

**Summary:**

This paper studies knowledge poisoning attacks on RAG systems, where the attacker injects malicious documents into the knowledge base to manipulate the system’s responses to specific queries. To improve the robustness of existing poisoning methods against defenses such as TrustRAG, the authors propose two techniques: concatenating malicious and benign documents, and optimizing the embedding distribution to disperse malicious documents in the embedding space, thereby evading cluster-based detection. Experiments show that the proposed approach achieves higher attack success rates against TrustRAG compared to baseline methods.

**Strengths:**

1. The paper is well-written and easy to follow. The proposed method is simple yet intuitive.
2. Experimental results validate the effectiveness of the approach against specific defenses, showing clear improvements over baseline methods.
3. The ablation study is comprehensive and clearly demonstrates the contribution of different components in the proposed design.

**Weaknesses:**

1. Among the two evaluated defenses, the paper introduces a query-detection defense that flags documents containing the query as malicious. However, adding queries to documents is a common IR technique for improving retrieval quality [1][2], making this defense questionable.
2. The proposed method appears tailored to specific defenses rather than offering a generalizable solution. Moreover, one of the evaluated defenses (query detection) may not be valid, as noted above. Other relevant defenses, such as Divide-and-Vote [3], are not considered.
3. The generated documents contain noticeable gibberish or unnatural phrases after perturbation and replacement (Table 8), which could make them easier to detect.
4. Although the method is designed to improve performance against TrustRAG, the attack success rate remains relatively low (~27%). Given this targeted improvement, stronger results would be expected. In other settings (Tables 3 and 4), performance is comparable to or worse than baselines.

**References**
[1] *Document Expansion by Query Prediction*, 2019.
[2] *Doc2Query--: When Less is More*, 2023.
[3] *On the Risk of Misinformation Pollution with Large Language Models*, 2023.

**Questions:**

1. How many adversarial (poisoned) documents are used for each target query? How does increasing this number affect the ASR and the defense detection rate? For example, when moving from 1-shot to multi-shot poisoning?

---

> ### Author Response · Authors · 2025-12-03
>
> ### **Compatibility of Query Detection with Document Expansion (Doc2Query)**
>
> We thank the reviewer for highlighting the potential interaction between the query-detection defense and document expansion techniques. We acknowledge that **Doc2Query** [1, 2] is a valuable IR technique where a Seq2Seq model predicts potential queries relevant to a document and appends them to the content to enhance retrieval.
>
> However, we argue that the query-detection defense, which we introduced as a baseline to evaluate attack robustness, remains valid and compatible with modern IR systems for the following reasons:
>
> **1. High Lexical Overlap vs. Semantic Prediction**
> The query-detection defense operates by identifying **high lexical overlap** between the user query and the retrieved content (specifically, we utilized a sliding window with a sequence matching threshold of 0.8 in our experiments). There is a fundamental distinction between this detection mechanism and the output of Doc2Query:
>
> * **Doc2Query:** Generates *predicted* queries. While these predictions are *semantically* relevant, they rarely align **lexically** with the specific phrasing of a user's real-time query (e.g., Doc2Query might predict "drug side effects" while the user asks "What are the adverse reactions of X?"). Consequently, the character-level similarity score typically remains well below the 0.8 threshold.
> * **Query Detection:** Targets the high lexical similarity (near-duplicates) characteristic of attacks like PoisonedRAG, which inject the specific target query string to maximize retrieval scores.
>
> **2. Security Necessity in Critical Domains**
> Even in scenarios where a conflict might exist (e.g., very short or common queries where Doc2Query might coincidentally match the user input), we argue that security must take precedence. In high-stakes domains such as healthcare or finance, the cost of a successful poisoning attack, which could lead to misinformation or financial loss, far outweighs the cost of a marginal drop in retrieval recall. If utilizing a specific IR technique like Doc2Query significantly increases the attack surface for techniques like PoisonedRAG, prioritizing a strict similarity-based filter is a necessary trade-off to ensure the system's safety and reliability.

---

> ### Author Response · Authors · 2025-12-03
>
> ### **Comparison with Recent Attack and Defense Baselines**
>
> Thank you for the insightful comment. We have expanded our evaluation to include the **CorruptRAG** [4] attack baseline and the **Divide-and-Vote** [3] defense baseline.
>
> **CorruptRAG** operates by injecting an adversarial document which claims that the corpus containing the correct answer is outdated, asserting the attacker’s target answer is the correct one.
> **Divide-and-Vote** [3] is a defense strategy that generates a prediction for each retrieved document individually and aggregates the results via majority voting.
>
> As shown in the table below, our experiments yielded the following insights:
>
> 1.  **Effectiveness against Query Detection:** CorruptRAG is ineffective against the Query Detection defense (ASR: 4.90%). This is because, similar to PoisonedRAG, CorruptRAG prepends the target query to the adversarial document to ensure retrieval. This specific characteristic makes it easily detectable by the query-detection mechanism.
>
> 2.  **Effectiveness against Divide-and-Vote:** The attack is also ineffective against Divide-and-Vote (ASR: 8.50%). This defense isolates the poisoned document from the benign documents by feeding them to the LLM separately. Since the LLM generates an output for each document in the top-k results, the single poisoned document affects only one prediction. The remaining benign documents produce correct answers, which dominate the aggregation result, leading to a correct final answer.
>
> 3.  **Effectiveness against TrustRAG:** CorruptRAG achieves a comparatively higher ASR against TrustRAG (34.80%). Because CorruptRAG injects only a single poisoned document, it does not form the dense clusters that TrustRAG's clustering defense relies on for detection. Furthermore, unlike PIA, it does not contain explicit malicious instructions, allowing it to bypass TrustRAG's malicious instruction filtering process.
>
> **Conclusion on Robustness:**
> While CorruptRAG achieves a higher ASR against TrustRAG specifically, it is rendered ineffective (<9% ASR) by the other two defenses. In contrast, CamoDocs demonstrates consistent effectiveness, maintaining a significant threat level (ASR > 29%) across *all* evaluated defenses. This confirms that CamoDocs is a more versatile attack strategy in diverse defense environments.
>
> | Defense | Attack | Attack Success Rate | Clean Accuracy |
> | :--- | :--- | :--- | :--- |
> | **Query Detection** | PoisonedRAG | 7.20 | 26.60 |
> | | PIA | 4.90 | 39.90 |
> | | CorruptRAG | 4.90 | 39.90 |
> | | CamoDocs | 68.90 | 15.40 |
> | **Divide-and-Vote** | PoisonedRAG | 53.50 | 12.00 |
> | | PIA | 8.00 | 27.50 |
> | | CorruptRAG | 8.50 | 28.00 |
> | | CamoDocs | 65.00 | 13.50 |
> | **TrustRAG** | PoisonedRAG | 6.50 | 33.30 |
> | | PIA | 7.60 | 39.20 |
> | | CorruptRAG | 34.80 | 37.90 |
> | | CamoDocs | 29.40 | 32.70 |
>
> [4] Zhang, B., Chen, Y., Fang, M., Liu, Z., Nie, L., Li, T., & Liu, Z. (2025). Practical poisoning attacks against retrieval-augmented generation. *arXiv preprint arXiv:2504.03957*.

---

> ### Author Response · Authors · 2025-12-03
>
> ### **On the Detectability of Unnatural Phrases**
>
> Thank you for the insightful comment. We agree that maintaining naturalness is important for stealth. However, we respectfully argue that detecting "unnatural" phrases in a real-world RAG pipeline is a non-trivial challenge that imposes severe trade-offs on the defender:
>
> 1.  **The False-Positive Dilemma in Real-World Data:** RAG systems frequently retrieve content from noisy sources (e.g., social media discussions, technical logs, OCR-scanned documents, or raw web scrapes). These benign documents inherently contain grammatical errors, slang, and fragmented text. A simple rule-based detector or aggressive perplexity filter would incorrectly flag valid information, significantly degrading the system's retrieval recall and utility.
> 2.  **Computational Overhead:** While a sophisticated LLM could detect subtle unnatural phrasing, deploying an LLM to scrutinize every retrieved document *before* generation introduces prohibitive latency and computational costs, rendering the RAG system impractical for real-time applications.
>
> **Improved Naturalness via Coherence Filtering:**
> Notwithstanding the difficulty of detection, we have further strengthened CamoDocs to address the reviewer's concern directly. We introduced a **Coherence Filter** (using a lightweight GPT-2 proxy) into our optimization loop. This ensures that selected token perturbations maintain low perplexity scores.
>
> **Implementation Details:**
> The coherence filter calculates the perplexity of the text resulting from candidate token replacements. We expanded the candidate pool size ($m$) in Algorithm 1 to 1,000. From this pool, we filtered candidates based on perplexity, retaining only the top 10 tokens that resulted in the most natural (lowest perplexity) text. We then evaluated the actual loss against the surrogate embedding model using only these 10 linguistically coherent candidates.
>
> **Results:**
> With this technique, we not only produced more natural phrases but also enhanced the Attack Success Rate (ASR) by up to **3.9 percentage points** with Llama-3.1-8B on HotpotQA against the TrustRAG defense. The qualitative improvement is demonstrated in the example below:
>
> > **Question:** Who was the defense counsel of a German woman who underwent Catholic exorcism rites during the year before her death?
> >
> > **Before Coherence Filter:** "Herories Witch with her Graves Weasley other Benefits which she took medication merry as Booster.\"\"Michelodder revival family additionally Prom she was possessed by demons.The drum attracted media and public attention …"
> >
> > **After Coherence Filter:** "Her condition worsened with greatness displaying multiple sad symptoms which she took history for as well?\"Michem and her family became convinced she was possessed following demons.The case spreading media and requested help due mainly the demons' unusual decision …"
>
> **Quantitative Performance with Coherence Filter against TrustRAG:**
>
> | Model | Method | Attack Success Rate | Clean Accuracy |
> | :--- | :--- | :--- | :--- |
> | **Mistral-Nemo (2407) 12.2B** | CamoDocs (Base) | 28.60 | 32.80 |
> | | **+ Coherence Filter** | **32.00** | 33.20 |
> | **Llama-3.1-8B** | CamoDocs (Base) | 29.40 | 32.70 |
> | | **+ Coherence Filter** | **33.30** | 31.30 |
> | **Mixtral-8x7B** | CamoDocs (Base) | 25.60 | 37.40 |
> | | **+ Coherence Filter** | **27.40** | 39.70 |

---

> ### Author Response · Authors · 2025-12-03
>
> ### **Justification for Attack Success Rate and Performance Comparison**
>
> Thank you for the insightful comment regarding the attack success rate (ASR) and performance comparisons. We address these points below:
>
> **1. Significance of 27% ASR against TrustRAG**
> In the field of AI security, particularly regarding adversarial attacks, the primary concern is vulnerability in critical domains such as healthcare and finance, where high reliability is mandatory. Even a single successful attack can have severe consequences. For instance, there is a real-world example where a retrieval corruption attack caused a $2.5K financial loss when an LLM generated code containing a malicious snippet.
>
> Consider a scenario where an LLM is deployed in the healthcare field; if the model provides incorrect medical information to doctors or patients due to a poisoning attack, the consequences could be catastrophic. Given these potential outcomes and the expected wide adoption of RAG systems, we argue that a 27.78% ASR, meaning more than 1 in 4 attacks succeed despite state-of-the-art defenses, represents an intolerable security risk.
>
> Furthermore, we demonstrated that this ASR is not a hard ceiling. By marginally increasing the poisoning ratio to 0.029% (which remains a negligible fraction of the corpus), CamoDocs achieved an ASR of 31.20% with Llama-3.1-8B on HotpotQA
>
> **2. Performance Compared to Baselines (Tables 3 and 4)**
> Regarding the observation that CamoDocs performs comparably to or slightly worse than baselines in undefended settings: This is an expected trade-off between stealth and raw effectiveness.
> * Baselines like **PoisonedRAG** and **PIA** achieve high ASR in no-defense settings by using conspicuous patterns (e.g., concatenating the exact target query or explicit malicious instructions). However, this makes them fragile; their ASR collapses when defenses like TrustRAG are applied.
> * **CamoDocs** is explicitly optimized to survive these defenses by dispersing embeddings and camouflaging content. While this constraint slightly limits its maximum potential in a completely undefended environment, it makes CamoDocs the only effective threat in realistic, defended scenarios where baselines are rendered ineffective.

---

> ### Author Response · Authors · 2025-12-03
>
> ### **Clarification on Poisoning Ratio and Its Impact on ASR**
>
> Thank you for the insightful and helpful comment.
>
> We used **10 poisoned documents per query** ($\beta = 10$) across all datasets. This is derived from setting **$k = 5$** for the top-k selection of benign documents and a chunk count $\gamma = 2$ ($5 \times 2 = 10$).
>
> As described in **Section 4.1**, we tested 50 queries per repetition. Consequently, the corresponding poisoning ratios are **0.01%**, **0.02%**, and **0.01%** for **HotpotQA**, **NQ**, and **MS-MARCO**, respectively. This poisoning ratio is much less than 1%, ensuring the true attack scale remains very small.
>
> The exact statistics are summarized in the table below. Note that the tested queries do not overlap between different repetitions. This means that testing 50 queries per repetition over 10 repetitions allows us to evaluate **500 queries total**, but with a reduced poisoning ratio. This reduced poisoning ratio occurs because we only inject adversarial documents for the target queries tested in the *current* repetition. This is distinct from a scenario where one tests 500 queries at once with all corresponding poisoned documents injected simultaneously. By injecting adversarial documents only for the 50 queries currently being tested, we maintain a realistic and low poisoning budget per experiment.
>
> | Dataset | Poisoned Docs per Query ($\beta$) | Tested Queries per Repetition | Total Poisoned Docs per Repetition | Corpus Size | Poisoning Ratio |
> | :--- | :--- | :--- | :--- | :--- | :--- |
> | HotpotQA | 10 | 50 | 500 | 5,233,329 | 0.01% |
> | NQ | 10 | 50 | 500 | 2,681,468 | 0.02% |
> | MS-MARCO | 10 | 50 | 500 | 8,841,823 | 0.01% |
>
> In the revised paper, we increased the number of tested queries per repetition from 50 to 100 as requested by **R-vpWp**. Consequently, the total poisoned documents per repetition and the poisoning ratio doubled from 0.009%, 0.019%, 0.006% to 0.019%, 0.037%, and 0.011%, respectively.
>
> Regarding the parameter $\beta$, we conducted an additional experiment to observe how it affects the attack success rate. We increased $\beta$ from 10 to 15, 20, and 25 by selecting the Top-K ($K=5$) retrieved documents using the surrogate retriever and setting the chunk count ($\gamma$) to 3, 4, and 5. In other words, we increased $\beta$ by splitting the benign and adversarial sub-documents in a more fine-grained manner. The results below were obtained with Llama-3.1-8B on HotpotQA under the TrustRAG defense.
>
> | $\beta$ | Attack Success Rate | Clean Accuracy | Poisoning Ratio | Proportion of Retrieved Adv. Docs (Before Defense) | Proportion of Retrieved Adv. Docs (After Defense) |
> | :--- | :--- | :--- | :--- | :--- | :--- |
> | 10 | 28.10 | 31.50 | 0.019% | 93.62% | 54.06% |
> | 15 | 31.20 | 32.10 | 0.029% | 93.40% | 65.42% |
> | 20 | 28.80 | 33.50 | 0.038% | 92.90% | 62.00% |
> | 25 | 20.20 | 33.00 | 0.048% | 95.38% | 46.66% |
>
> As seen in the table above, increasing $\beta$ and the poisoning ratio helps increase the attack success rate to some extent; for instance, the attack success rate increases by 3.1 percentage points when $\beta$ increases from 10 to 15. However, the attack success rate does not increase monotonically with $\beta$, as it decreases again starting from $\beta=20$.
>
> This is because, although the increased poisoning ratio helps increase the proportion of retrieved adversarial documents among the top-k retrieved documents, it also aids the K-means clustering defense in detecting them. A large number of retrieved adversarial documents tends to form a tight cluster that is easily detectable by the K-means algorithm. Conversely, a small number of retrieved adversarial documents is less detectable because there are fewer points to form a cluster. This analysis is confirmed by the steep decline in the proportion of retrieved adversarial documents **after** applying the K-means clustering defense as $\beta$ increases to 25 (dropping to 46.66%), even though the proportion **before** defense increases (95.38%).

---

### Official Review · Reviewer_42uV · 2025-11-01

**Soundness:** 2
**Presentation:** 2
**Contribution:** 2
**Rating:** 4
**Confidence:** 3

**Summary:**

This paper presents CamoDocs, a poisoning attack on RAG systems that hides malicious content inside normal-looking documents. It blends adversarial and benign text, then tweaks the benign parts so the documents evade detection. Tested on Llama-3, Mixtral, and Mistral across QA benchmarks, CamoDocs achieves up to 70% attack success and still 27% under defenses like TrustRAG. The study warns that RAG pipelines are highly vulnerable to stealthy data poisoning and calls for stronger defenses.

**Strengths:**

1. This paper gives a concrete two-stage method (chunking + token manipulation) with an algorithmic description.
2. It includes embedding visualizations and distance/KDE analyses to explain why the attack evades clustering defenses.
3. This paper assumes black-box access to LLMs/retrievers and only ability to inject documents, matching real-world constraints.

**Weaknesses:**

1. The paper does not provide any evaluation of the runtime or computational cost of the proposed CamoDocs attack or its baselines. Since the method involves multi-stage operations: document chunking, iterative token manipulation, and surrogate model optimization. Understanding runtime overhead is essential. Without these measurements, it is difficult to assess whether the attack is feasible on large-scale real-world RAG systems or only in small controlled experiments.
2.  The paper mentions a total poisoning ratio of less than 1% but does not explicitly report how many poisoned documents are injected per query or how the parameter β (the target number of adversarial documents) is chosen. This omission makes it difficult to reproduce the experiments or to assess the true attack scale and stealthiness. Without a clear specification of per-query injection count, readers cannot determine whether the reported success rates are achievable under realistic constraints or depend on an unrealistically large poisoning budget.
3. The defense side is restricted to simple heuristic mechanisms—TrustRAG, query detection, query rephrasing, and perplexity filtering. These methods are either rule-based or heuristic, lacking consideration of adaptive or learning-based defense strategies such as robust retriever training, certified filtering, or contrastive anomaly detection. Consequently, the defense evaluation may underestimate how current or future systems could mitigate such attacks.
4. While CamoDocs empirically achieves high attack success rates, the paper lacks a theoretical or analytical explanation of why dispersed embeddings and mixed sub-documents so effectively bypass clustering-based defenses. A more formal discussion could clarify whether this behavior is dataset-specific or reflects a general weakness in embedding-space defenses.

**Questions:**

N/A

---

> ### Author Response · Authors · 2025-12-03
>
> ## **Computational cost analysis**
>
> Thank you for the insightful comment. We conducted a computational cost analysis for each stage of our algorithm. We synthesize adversarial documents with the synthesizer LLM (GPT-4o-mini), which is also used in PoisonedRAG, and we use a lightweight word-based sparse retriever (BM25) as a surrogate retriever.
>
> ### **Computational cost for document chunking**
>
> Chunking a document requires a single pass over the document. Let the average document length for benign documents be $L_{bn}$, the average document length for adversarial documents be $L_{adv}$, and let $k$ be the number of retrieved benign documents and synthesized adversarial documents per query.
>
> Then, the computational cost for document chunking is $O(kL_{bn} + kL_{adv})$, since we need one pass over each text for chunking.
>
> ### **Computational cost for token manipulation**
>
> Token manipulation requires one batched forward pass with batch size equal to the final target number of adversarial documents $\beta$, one batched backward pass with the loss $L$ to obtain the gradient $\nabla_{e_t} L$ used for the first-order approximation, and $m$ additional forward passes, where $m$ is the number of candidate tokens that show high approximate loss. After chunking, each document is split into $\gamma$ sub-documents, so the average sub-document length becomes $L_{\text{sub, bn}} = L_{bn} / \gamma$ and $L_{\text{sub, adv}} = L_{adv} / \gamma$ for benign and adversarial sub-documents, respectively.
>
> Let $T_{\text{fwd}}(L_{\text{sub}})$ and $T_{\text{bwd}}(L_{\text{sub}})$ denote the time for a single forward pass and a single backward pass, respectively, of the surrogate encoder $E_{\text{surr}}$ on a sub-document of length $L_{\text{sub}}$. Under the standard assumption that encoder cost scales linearly with batch size, one batched forward pass over the $\beta$ current benign sub-documents costs $\beta T_{\text{fwd}}(L_{\text{sub,bn}})$ and the corresponding batched backward pass costs $\beta T_{\text{bwd}}(L_{\text{sub,bn}})$.
>
> In addition, given the gradient with respect to the chosen token’s embedding, we score at most $m$ candidate token embeddings using inner products $\nabla_{e_t} L \cdot e_t$, which costs $O(md)$ per step, where $d$ is the embedding dimension.
>
> The $m$ candidate evaluations require $m T_{\text{fwd}}(L_{\text{sub,bn}})$.
>
> Therefore, each token-manipulation step has complexity
> $$
> O\big(\beta T_{\text{fwd}}(L_{\text{sub,bn}}) + \beta T_{\text{bwd}}(L_{\text{sub,bn}}) + m T_{\text{fwd}}(L_{\text{sub,bn}}) + md\big),
> $$
> and optimizing for $\alpha$ steps yields a total token-manipulation cost per query of
> $$
> O\big(\alpha [\beta T_{\text{fwd}}(L_{\text{sub,bn}}) + \beta T_{\text{bwd}}(L_{\text{sub,bn}}) + m T_{\text{fwd}}(L_{\text{sub,bn}}) + md]\big).
> $$
>
> Therefore, the overall cost is dominated by the encoder forward and backward passes.
>
> ### **Computational cost for sub-document merging**
>
> After token manipulation, we merge the optimized benign sub-documents with the adversarial sub-documents to form the final adversarial documents.
>
> To form one final adversarial document, we read one optimized benign sub-document of length $L_{bn}/\gamma$
> and one adversarial sub-document of length $L_{adv}/\gamma$, and we write their concatenation, which has length $L_{bn}/\gamma + L_{adv}/\gamma$.
>
> Thus, the total number of token operations (reads and writes) per adversarial document is $2[L_{bn}/\gamma + L_{adv}/\gamma]$.
> Repeating this process for the $\beta$ final adversarial documents per query yields a total sub-document merging cost of $O\big(\beta \cdot [2(L_{bn}/\gamma + L_{adv}/\gamma)]\big)$.
>
> ### **Overall per-query complexity**
>
> Putting these stages together, the computational cost of CamoDocs for a single target query consists of:
>
> - document chunking: $O(kL_{bn} + kL_{adv})$,
> - token manipulation: $O\big(\alpha [\beta T_{\text{fwd}}(L_{\text{sub,bn}}) + \beta T_{\text{bwd}}(L_{\text{sub,bn}}) + m T_{\text{fwd}}(L_{\text{sub,bn}}) + md]\big)$,
> - sub-document merging: $O\big(\beta \cdot [2(L_{bn}/\gamma + L_{adv}/\gamma)]\big)$.
>
> All of these costs are incurred **offline** when constructing the poisoned documents. At inference time, the RAG system with poisoned documents uses the same retriever and LLM as in the clean setting and PoisonedRAG, so the online latency is essentially unchanged.
>
> To generate one adversarial document for one target query, CamoDocs takes about **32 seconds** on a single A6000 GPU with PyTorch 2.7.1 and CUDA 12.6. Token manipulation dominates the runtime because it requires multiple forward passes and one batched backward pass of the embedding model. We randomly selected 100 HotpotQA test queries, measured the time for each procedure, and report the averages.
>
> **Breakdown of CamoDocs runtime**
>
> |Procedure|Average Runtime (seconds)|
> |-|-|
> |Document chunking|0.000|
> |Token manipulation|31.754|
> |Sub-document merging|0.000|

---

> ### Author Response · Authors · 2025-12-03
>
> ### **Clarification on Poisoning Ratio and Injection Parameters ($\beta$)**
>
> Thank you for the insightful and helpful comment.
>
> We used **10 poisoned documents per query** ($\beta = 10$) across all datasets. This is derived from setting **$k = 5$** for the top-k selection of benign documents and a chunk count $\gamma = 2$ ($5 \times 2 = 10$).
>
> As described in **Section 4.1**, we tested 50 queries per repetition. Consequently, the corresponding poisoning ratios are **0.01%**, **0.02%**, and **0.01%** for **HotpotQA**, **NQ**, and **MS-MARCO**, respectively. This poisoning ratio is much less than 1%, ensuring the true attack scale remains very small.
>
> The exact statistics are summarized in the table below. Note that the tested queries do not overlap between different repetitions. This means that testing 50 queries per repetition over 10 repetitions allows us to evaluate **500 queries total**, but with a reduced poisoning ratio. This reduced poisoning ratio occurs because we only inject adversarial documents for the target queries tested in the *current* repetition. This is distinct from a scenario where one tests 500 queries at once with all corresponding poisoned documents injected simultaneously. By injecting adversarial documents only for the 50 queries currently being tested, we maintain a realistic and low poisoning budget per experiment.
>
> | Dataset | Poisoned Docs per Query ($\beta$) | Tested Queries per Repetition | Total Poisoned Docs per Repetition | Corpus Size | Poisoning Ratio |
> | :--- | :--- | :--- | :--- | :--- | :--- |
> | HotpotQA | 10 | 50 | 500 | 5,233,329 | 0.01% |
> | NQ | 10 | 50 | 500 | 2,681,468 | 0.02% |
> | MS-MARCO | 10 | 50 | 500 | 8,841,823 | 0.01% |
>
> In the revised paper, we increased the number of tested queries per repetition from 50 to 100 as requested by **R-vpWp**. Consequently, the total poisoned documents per repetition and the poisoning ratio doubled from 0.009%, 0.019%, 0.006% to 0.019%, 0.037%, and 0.011%, respectively.
>
> Regarding the parameter $\beta$, we conducted an additional experiment to observe how it affects the attack success rate. We increased $\beta$ from 10 to 15, 20, and 25 by selecting the Top-K ($K=5$) retrieved documents using the surrogate retriever and setting the chunk count ($\gamma$) to 3, 4, and 5. In other words, we increased $\beta$ by splitting the benign and adversarial sub-documents in a more fine-grained manner. The results below were obtained with Llama-3.1-8B on HotpotQA under the TrustRAG defense.
>
> | $\beta$ | Attack Success Rate | Clean Accuracy | Poisoning Ratio | Proportion of Retrieved Adv. Docs (Before Defense) | Proportion of Retrieved Adv. Docs (After Defense) |
> | :--- | :--- | :--- | :--- | :--- | :--- |
> | 10 | 28.10 | 31.50 | 0.019% | 93.62% | 54.06% |
> | 15 | 31.20 | 32.10 | 0.029% | 93.40% | 65.42% |
> | 20 | 28.80 | 33.50 | 0.038% | 92.90% | 62.00% |
> | 25 | 20.20 | 33.00 | 0.048% | 95.38% | 46.66% |
>
> As seen in the table above, increasing $\beta$ and the poisoning ratio helps increase the attack success rate to some extent; for instance, the attack success rate increases by 3.1 percentage points when $\beta$ increases from 10 to 15. However, the attack success rate does not increase monotonically with $\beta$, as it decreases again starting from $\beta=20$.
>
> This is because, although the increased poisoning ratio helps increase the proportion of retrieved adversarial documents among the top-k retrieved documents, it also aids the K-means clustering defense in detecting them. A large number of retrieved adversarial documents tends to form a tight cluster that is easily detectable by the K-means algorithm. Conversely, a small number of retrieved adversarial documents is less detectable because there are fewer points to form a cluster. This analysis is confirmed by the steep decline in the proportion of retrieved adversarial documents **after** applying the K-means clustering defense as $\beta$ increases to 25 (dropping to 46.66%), even though the proportion **before** defense increases (95.38%).

---

> ### Author Response · Authors · 2025-12-03
>
> ### **Comparison with Recent Attack and Defense Baselines**
>
> Thank you for the insightful comment. We have expanded our evaluation to include the **CorruptRAG** [1] attack baseline and the **Divide-and-Vote** [2] defense baseline.
>
> **CorruptRAG** operates by injecting an adversarial document which claims that the corpus containing the correct answer is outdated, asserting the attacker’s target answer is the correct one.
> **Divide-and-Vote** [2] is a defense strategy that generates a prediction for each retrieved document individually and aggregates the results via majority voting.
>
> As shown in the table below, our experiments yielded the following insights:
>
> 1.  **Effectiveness against Query Detection:** CorruptRAG is ineffective against the Query Detection defense (ASR: 4.90%). This is because, similar to PoisonedRAG, CorruptRAG prepends the target query to the adversarial document to ensure retrieval. This specific characteristic makes it easily detectable by the query-detection mechanism.
>
> 2.  **Effectiveness against Divide-and-Vote:** The attack is also ineffective against Divide-and-Vote (ASR: 8.50%). This defense isolates the poisoned document from the benign documents by feeding them to the LLM separately. Since the LLM generates an output for each document in the top-k results, the single poisoned document affects only one prediction. The remaining benign documents produce correct answers, which dominate the aggregation result, leading to a correct final answer.
>
> 3.  **Effectiveness against TrustRAG:** CorruptRAG achieves a comparatively higher ASR against TrustRAG (34.80%). Because CorruptRAG injects only a single poisoned document, it does not form the dense clusters that TrustRAG's clustering defense relies on for detection. Furthermore, unlike PIA, it does not contain explicit malicious instructions, allowing it to bypass TrustRAG's malicious instruction filtering process.
>
> **Conclusion on Robustness:**
> While CorruptRAG achieves a higher ASR against TrustRAG specifically, it is rendered ineffective (<9% ASR) by the other two defenses. In contrast, CamoDocs demonstrates consistent effectiveness, maintaining a significant threat level (ASR > 29%) across *all* evaluated defenses. This confirms that CamoDocs is a more versatile attack strategy in diverse defense environments.
>
> | Defense | Attack | Attack Success Rate | Clean Accuracy |
> | :--- | :--- | :--- | :--- |
> | **Query Detection** | PoisonedRAG | 7.20 | 26.60 |
> | | PIA | 4.90 | 39.90 |
> | | CorruptRAG | 4.90 | 39.90 |
> | | CamoDocs | 68.90 | 15.40 |
> | **Divide-and-Vote** | PoisonedRAG | 53.50 | 12.00 |
> | | PIA | 8.00 | 27.50 |
> | | CorruptRAG | 8.50 | 28.00 |
> | | CamoDocs | 65.00 | 13.50 |
> | **TrustRAG** | PoisonedRAG | 6.50 | 33.30 |
> | | PIA | 7.60 | 39.20 |
> | | CorruptRAG | 34.80 | 37.90 |
> | | CamoDocs | 29.40 | 32.70 |
>
> [1] Zhang, B., Chen, Y., Fang, M., Liu, Z., Nie, L., Li, T., & Liu, Z. (2025). Practical poisoning attacks against retrieval-augmented generation. *arXiv preprint arXiv:2504.03957*.
>
> [2] Pan, Y., Pan, L., Chen, W., Nakov, P., Kan, M. Y., & Wang, W. (2023). On the Risk of Misinformation Pollution with Large Language Models. *Findings of the Association for Computational Linguistics: EMNLP 2023*.

---

> ### Author Response · Authors · 2025-12-03
>
> ### **An Analytical Explanation for CamoDocs**
>
> Thank you for the insightful comment. The loss function of CamoDocs is designed to exploit a general weakness in embedding-space defenses. Here, we present an analytical explanation for this mechanism.
>
> Embedding-space defenses, such as TrustRAG, operate on the assumption that the embeddings of adversarial documents form an anomalously tight cluster (low variance). Conversely, our loss function is designed to maximize the trace of the sample covariance matrix of the adversarial document embeddings. Maximizing the trace corresponds to maximizing the variance across all embedding dimensions, which inflates the distribution of the adversarial embeddings and effectively violates the underlying assumption of these defenses.
>
> Recall that the loss function of CamoDocs is defined as:
>
> $$
> \mathcal{L} = \frac{1}{\beta}\sum_{j=1}^\beta \|| e_{q_i,j} - c \||_2
> $$
>
> where $e_{q_i,j} \in \mathbb{R}^{h \times 1}$ is the embedding of the $j$-th optimized benign sub-document, $h$ is the hidden dimension of the embedding model, and $c = \frac{1}{\beta}\sum_{j=1}^\beta e_{q_i,j}$ denotes the centroid.
>
> We demonstrate that maximizing the Euclidean distance $\|| e_{q_i,j} - c \||_2$ is analytically equivalent to maximizing the trace of the sample covariance matrix, $\text{Tr}(\hat{\Sigma})$. The sample covariance matrix is given by:
>
> $$
> \hat{\Sigma} = \frac{1}{\beta-1} \sum_{j=1}^{\beta} (e_{q_i,j} - c)(e_{q_i,j} - c)^T
> $$
>
> Using the properties of the trace operator, we derive:
>
> $$
> \begin{aligned}
> \text{Tr}(\hat{\Sigma}) &= \frac{1}{\beta - 1} \sum_{j=1}^\beta \text{Tr}\left( (e_{q_i,j} - c)(e_{q_i,j} - c)^T \right) && (\text{Linearity of the trace operation}) \newline
> &= \frac{1}{\beta - 1} \sum_{j=1}^\beta \text{Tr}\left( (e_{q_i,j} - c)^T(e_{q_i,j} - c) \right) && (\text{Cyclic property of the trace operation}) \newline
> &= \frac{1}{\beta - 1} \sum_{j=1}^\beta (e_{q_i,j} - c)^T(e_{q_i,j} - c) && (\text{Trace of a scalar is the scalar itself}) \newline
> &= \frac{1}{\beta - 1} \sum_{j=1}^\beta \|| e_{q_i,j} - c \||_2^2
> \end{aligned}
> $$
>
> The primary distinction between the CamoDocs loss $\mathcal{L}$ and $\text{Tr}(\hat{\Sigma})$ is that $\mathcal{L}$ utilizes the $L_2$ norm inside the summation, whereas the trace corresponds to the **squared** $L_2$ norm.
>
> However, the gradient of the norm, $\nabla_x \||x\||_2 = \frac{x}{\||x\||_2}$, and the gradient of the squared norm, $\nabla_x \||x\||_2^2 = 2x$, point in the exact same direction (radially outward from the centroid), differing only in magnitude.
>
> Therefore, by performing gradient ascent on $\mathcal{L}$, CamoDocs implicitly maximizes $\text{Tr}(\hat{\Sigma})$, which represents the sum of the variances of the embedding along all dimensions, as $\text{Tr}(\hat{\Sigma}) \propto \sum \|| e_{q_i,j} - c \||_2^2$. This process forces the adversarial embeddings to disperse along all dimensions, thereby breaking the density assumption upon which embedding-space defenses rely.

---

### Author Response · Authors · 2025-12-04
**General Response to All Reviewers**

We sincerely thank all the reviewers for their insightful comments. We have provided detailed responses to each reviewer below and have also uploaded a revised version of our submission.

---

### Meta-Review · Area_Chair_92wm · 2026-01-12

**Summary:**

The overall evaluation leans toward rejection, with all reviewers recommending rejection. Reviewers acknowledge that the paper presents a clear, well-motivated poisoning attack on RAG systems with a concrete two-stage design, strong empirical evaluation across multiple models and datasets, and insightful analyses (e.g., embedding dispersion, ablations). However, the paper was initially criticized for missing critical experimental details (runtime cost, poisoning scale), limited and heuristic defense comparisons, unclear experimental design (number of queries, poisoning ratio), and concerns about generality, realism, and overstatement of impact. While the rebuttal substantially strengthens the paper by addressing many factual and experimental gaps, remaining concerns about generality of defenses, reliance on heuristic baselines, and modest ASR under strong defenses prevent a clear acceptance recommendation.

**Reviewer Concerns:**

**The rebuttal was submitted in December, and consequently no reviewers participated in the discussion.**

The authors provide a detailed and convincing computational cost analysis, showing that the attack is feasible offline with quantified runtime overhead. They clearly specify the number of poisoned documents per query, poisoning ratios, and experimental protocol, resolving reproducibility and scale ambiguities. Additional experiments significantly strengthen the evaluation: comparisons with recent attack and defense baselines (e.g., CorruptRAG, Divide-and-Vote), sensitivity analysis to surrogate–victim retriever mismatch, experiments on a stronger SOTA retriever, and evaluation against DBSCAN-style density-based defenses. The authors also add an analytical explanation linking the loss function to variance maximization in embedding space, and introduce a coherence filter to address concerns about unnatural or gibberish text. Finally, the clarification on the number of evaluated queries and the expanded sample size substantially improves confidence in the empirical results.


Despite these improvements, several concerns remain. Some reviewers remain unconvinced that the evaluated defenses adequately represent realistic, adaptive, or learning-based protection mechanisms deployed in modern RAG systems. While additional baselines were added, the defense landscape is still limited, and broader conclusions about general robustness may be premature. The attack success rate against TrustRAG, although argued to be “intolerable,” remains relatively modest, and the framing may still feel overstated to some readers. Questions also remain about whether the proposed embedding-dispersion strategy would generalize to future retrievers or more sophisticated anomaly detection pipelines beyond those tested. Finally, although the rebuttal addresses many weaknesses, the core contribution is still viewed by some reviewers as incremental rather than fundamentally novel.

**Reviewer Scores:**

**The rebuttal was submitted in December, and consequently no reviewers participated in the discussion.**

Reviewer 42uV (original rating: 4): Likely unchanged or slightly improved, given that runtime, poisoning scale, theory, and defense comparisons were all directly addressed.

Reviewer uH6r (original rating: 2): Likely improved modestly (e.g., to borderline reject), as key questions about poisoning scale, Divide-and-Vote defense, and query-detection validity were explicitly answered.

Reviewer vpWp (original rating: 2, high confidence): likely remains negative, as concerns about defense breadth and overall contribution persist despite additional experiments.

Reviewer e2hp (original rating: 4): Likely unchanged, although the added experiments mitigate some technical concerns, the reviewer’s skepticism regarding the paper’s core claims is unlikely to be fully alleviated.

---

### Decision · Program_Chairs · 2026-01-26

Reject